# Sonic hedgehog signaling directs patterned cell remodeling during cranial neural tube closure

Eric R Brooks[1], Mohammed Tarek Islam[1], Kathryn V Anderson[2], Jennifer A Zallen[1]*

[1]Howard Hughes Medical Institute and Developmental Biology Program, Sloan Kettering Institute, New York, United States; [2]Developmental Biology Program, Sloan Kettering Institute, New York, United States

**Abstract** Neural tube closure defects are a major cause of infant mortality, with exencephaly accounting for nearly one-third of cases. However, the mechanisms of cranial neural tube closure are not well understood. Here, we show that this process involves a tissue-wide pattern of apical constriction controlled by Sonic hedgehog (Shh) signaling. Midline cells in the mouse midbrain neuroepithelium are flat with large apical surfaces, whereas lateral cells are taller and undergo synchronous apical constriction, driving neural fold elevation. Embryos lacking the Shh effector Gli2 fail to produce appropriate midline cell architecture, whereas embryos with expanded Shh signaling, including the IFT-A complex mutants *Ift122* and *Ttc21b* and embryos expressing activated Smoothened, display apical constriction defects in lateral cells. Disruption of lateral, but not midline, cell remodeling results in exencephaly. These results reveal a morphogenetic program of patterned apical constriction governed by Shh signaling that generates structural changes in the developing mammalian brain.

*For correspondence:
zallenj@mskcc.org

Competing interests: The authors declare that no competing interests exist.

## Introduction

Neural tube closure defects are among the most common structural birth defects, occurring in 1 in 1000 pregnancies worldwide (*Wallingford et al., 2013*; *Zaganjor et al., 2016*). During development, neuroepithelial cells undergo extensive remodeling to transform a flat sheet into a fully closed tube that gives rise to the brain and spinal cord of the animal. Distinct genetic circuits are required for neural tube closure in different regions along the head-to-tail axis, translating positional information into location-appropriate cell behaviors (*Wilde et al., 2014*; *Aw and Devenport, 2017*; *Nikolopoulou et al., 2017*; *Juriloff and Harris, 2018*). Although many studies have focused on mechanisms of neural tube closure in the spinal cord, one-third of human neural tube defects arise from a failure of closure in the cranial region, resulting in exencephaly—an inoperable and terminally lethal birth defect (*Zaganjor et al., 2016*). More than a hundred genes are specifically required for closure of the mouse cranial neural plate, suggesting that unique mechanisms promote neural tube closure in the cranial region (*Harris and Juriloff, 2007*; *Harris and Juriloff, 2010*; *Wilde et al., 2014*). Despite the clinical importance of this disease, the cellular mechanisms that produce cranial neural tube structure, and how these cell behaviors are coordinated across thousands of cells to close the massive cranial region, remain opaque.

Tissue-scale structural changes during cranial neural closure require the precise spatial regulation of cell behaviors along the anterior-posterior and mediolateral axes. However, how cell behaviors are dynamically patterned along these axes is only beginning to be understood. The neural plate is significantly wider in the cranial region compared with the spinal cord, suggesting that distinct strategies are required for closure of the developing brain. In addition, positionally regulated signals

produce distinct cell fates along the mediolateral axis of the neural tube. Neuronal identities at different mediolateral positions are regulated by the secreted Shh, Wnt, and BMP proteins, with high levels of Shh producing ventral cell fates, moderate levels of Shh producing intermediate cell fates, and high levels of Wnt and BMP producing dorsal cell fates (*McMahon et al., 2003*; *Dessaud et al., 2008*; *Sagner and Briscoe, 2019*). In the posterior spinal cord, spatially restricted Shh and BMP signaling are required for local tissue bending, suggesting that these signals can influence tissue structure as well as cell identity (*Ybot-Gonzalez et al., 2002*; *Ybot-Gonzalez et al., 2007*). However, the cell behaviors that drive cranial neural tube closure and the positional signals that determine where and when these behaviors occur in the tissue are unknown.

Midline cells are essential drivers of neural tube closure across the chordate lineage, undergoing cell-shape changes (*Burnside and Jacobson, 1968*; *Smith et al., 1994*; *Haigo et al., 2003*; *Lee et al., 2007*; *Nishimura and Takeichi, 2008*; *Nishimura et al., 2012*; *McShane et al., 2015*) and planar rearrangements (*Davidson and Keller, 1999*; *Wallingford and Harland, 2002*; *Williams et al., 2014*; *Sutherland et al., 2020*) that narrow and bend the neural plate. At later stages of closure, cells at the borders of the neural plate form dynamic protrusions and adhesions that join the left and right sides of the neural plate to produce a closed tube (*Pyrgaki et al., 2010*; *Massarwa et al., 2014*; *Hashimoto et al., 2015*; *Ray and Niswander, 2016a*; *Ray and Niswander, 2016b*; *Molè et al., 2020*). However, it is not known if localized forces at the midline and borders of the neural plate are sufficient for closure of the significantly larger cranial region, or if distinct cell populations and behaviors contribute to cranial neural structure.

Apical constriction is a highly conserved process that transforms columnar epithelial cells into wedge shapes through actomyosin-dependent contraction of the apical cell surface and drives structural changes such as cell ingression, tissue bending, and tissue invagination (*Martin and Goldstein, 2014*). In the amphibian neural plate, apical constriction is required to form the median and dorsolateral hinge points, two localized tissue bending events that are a prerequisite for closure (*Burnside and Jacobson, 1968*; *Burnside, 1973*; *Haigo et al., 2003*; *Lee et al., 2007*; *Itoh et al., 2014*; *Ossipova et al., 2014*). However, it is not known if apical constriction contributes to closure in the tightly packed, pseudostratified neuroepithelium of the mammalian neural plate. In the mouse spinal cord, neural tube closure is independent of actomyosin activity, suggesting that apical constriction is not required for this process (*Ybot-Gonzalez and Copp, 1999*; *Escuin et al., 2015*). Instead, bending of the developing spinal cord is proposed to occur through alternative mechanisms such as tissue buckling or cell-cycle-dependent changes in nuclear position (*McShane et al., 2015*; *Nikolopoulou et al., 2017*). By contrast, regulators of actin and myosin are required for closure of the cranial neural plate, although the cell behaviors that are controlled by this contractile machinery are unclear (*Morriss-Kay and Tuckett, 1985*; *Hildebrand and Soriano, 1999*; *Brouns et al., 2000*; *McGreevy et al., 2015*). Loss of the actomyosin regulator Shroom3 leads to an increase in apical cell surface area in the cranial neuroepithelium, consistent with a defect in apical constriction (*McGreevy et al., 2015*). However, mammalian cranial neuroepithelial cells also undergo significant elongation along the apicobasal axis that can decrease the apical surface of cells independently of apical constriction (*Jacobson and Tam, 1982*), and several mutants defective for apicobasal elongation, including *Pten, Cfl1*, and *Nuak1/2* mutants, also show an increase in apical cell area (*Ohmura et al., 2012*; *Grego-Bessa et al., 2015*; *Grego-Bessa et al., 2016*). Disambiguating the contributions of apical constriction and apicobasal elongation to cranial closure is challenging, in part due to the difficulty in visualizing individual cell shapes in this densely packed tissue. Therefore, the cell behaviors that promote cranial neural closure, and the critical force-generating cell populations that drive these dynamic changes, are unknown.

Using high-resolution imaging of cell behavior in the mouse cranial neural plate, we demonstrate a tissue-wide pattern of apical constriction during neural tube closure in the developing midbrain. In contrast to the spinal cord, elevation of the cranial neural folds is driven by the synchronous, sustained apical constriction of a large population of lateral cells, whereas midline cells remain flat and apically expanded. The loss of Gli2, a transcriptional effector of Shh signaling, disrupts cell architecture at the midline, whereas loss of the IFT-A complex components Ift122 or Ttc21b disrupt apical constriction and actomyosin organization in lateral cells, resulting in a failure of cranial neural tube closure. These apical remodeling defects are recapitulated by activation of the Shh response throughout the midbrain, indicating that they are due to deregulated Shh signaling. Together, these

results demonstrate that lateral cells drive cranial neural tube closure through large-scale, coordinated apical constriction behaviors that are spatially regulated by patterned Shh activity.

## Results

### Neuroepithelial cells display patterned apical constriction during cranial closure

A critical step in the closure of the mouse midbrain is the transformation of the neural plate from convex to concave (*Figure 1A–C*; *Nikolopoulou et al., 2017*; *Vijayraghavan and Davidson, 2017*; *Juriloff and Harris, 2018*). Prior to closure, the cranial neural plate has an open, rams-horn shape (*Figure 1C*). The neuroepithelial sheet is convex on either side of the midline, with the outer edges of the neural plate tucked under the lateral regions. This curvature reverses during neural fold elevation, when both sides of the neural plate rise up and straighten to produce a concave, V-shaped structure (*Figure 1C*). The borders of the neural plate subsequently bend inward, appose, and fuse at the dorsal midline to produce a closed tube. To investigate the cell behaviors that drive these structural changes, we used confocal imaging and semi-automated image segmentation (*Mashburn et al., 2012*; *Farrell et al., 2017*) to analyze cell behavior at single-cell resolution. The apical profiles of midbrain neuroepithelial cells were relatively homogeneous in area prior to elevation (0 somites, E7.75) (*Figure 1D and E*). However, a strong pattern emerged during elevation (6 somites, E8.5). Lateral cells on either side of the midline displayed a more than 50% decrease in apical area between 0 and 9 somites (*Figure 1F–H*, *Supplementary file 1*). By contrast, the average apical surface area of midline cells did not change significantly during elevation (*Figure 2A–C*). Additionally, lateral cells became progressively mediolaterally oriented during the same period, whereas midline cell orientation was unchanged (*Figure 2—figure supplement 1*). These results indicate that lateral cells, but not midline cells, undergo apical remodeling during cranial neural fold elevation.

The finding that midline cells in the midbrain do not remodel during neural fold elevation differs from neural tube closure mechanisms in the spinal cord, in which wedge-shaped midline cells drive tissue bending (*McShane et al., 2015*; *Schoenwolf and Franks, 1984*; *Smith et al., 1994*; *Smith and Schoenwolf, 1988*), and raises the possibility that lateral cells may be key drivers of elevation. To determine if the apical remodeling of lateral cells is due to apicobasal elongation, we analyzed cell height in the cranial neural plate at different stages of elevation. Cell height in the lateral and midline regions did not change significantly during early elevation (0–7 somites) (*Figure 3C and D*), even though the average apical area of lateral cells decreased by more than 30% during this period (*Figure 1G*). By contrast, lateral and midline cells elongated more than 60% along the apical-basal axis after the 7-somite stage, such that lateral cells were consistently taller than midline cells throughout elevation (*Figure 3C–E*). Thus, cell remodeling in the elevating midbrain occurs in two phases, with an early phase involving apical remodeling in the absence of changes in cell height, and a later phase involving apicobasal elongation (*Figure 3J*). These results indicate that apicobasal elongation in the neuroepithelium occurs at late stages of elevation, but cannot explain the dramatic structural changes that occur during early elevation.

We next investigated whether apical constriction contributes to early structural changes in the midbrain neuroepithelium. Consistent with this possibility, the conversion of the midbrain neural plate from convex to concave is accompanied by a decrease in the apical span of the tissue without a significant change in the basal span (*Figure 3A and B*, *Figure 3—figure supplement 1A and B*). However, apical constriction has not been directly observed in the pseudostratified mammalian neural plate, where the crowded packing of cells has been proposed to hinder this process (*Nikolopoulou et al., 2017*). To determine if lateral cells undergo apical constriction, we visualized cell morphology in the midbrain neuroepithelium of embryos expressing membrane-GFP in a mosaic pattern, using the inefficient EIIA-Cre recombinase to label individual cells (*Figure 3F*; *Lakso et al., 1996*; *Muzumdar et al., 2007*). Using this approach, we identified five classes of lateral cells (*Figure 3G*). More than half of lateral cells (51 ± 6%) had a highly constricted apical neck, a hallmark of apical constriction. An additional 13 ± 2% of cells displayed properties consistent with apical constriction, but with a shorter neck domain, suggesting that apical area changes can occur even in the absence of a basal shift in cell volume. The remaining one-third of lateral cells were apically expanded, spindle-shaped, or columnar. Because few midline cells were labeled by this method, we

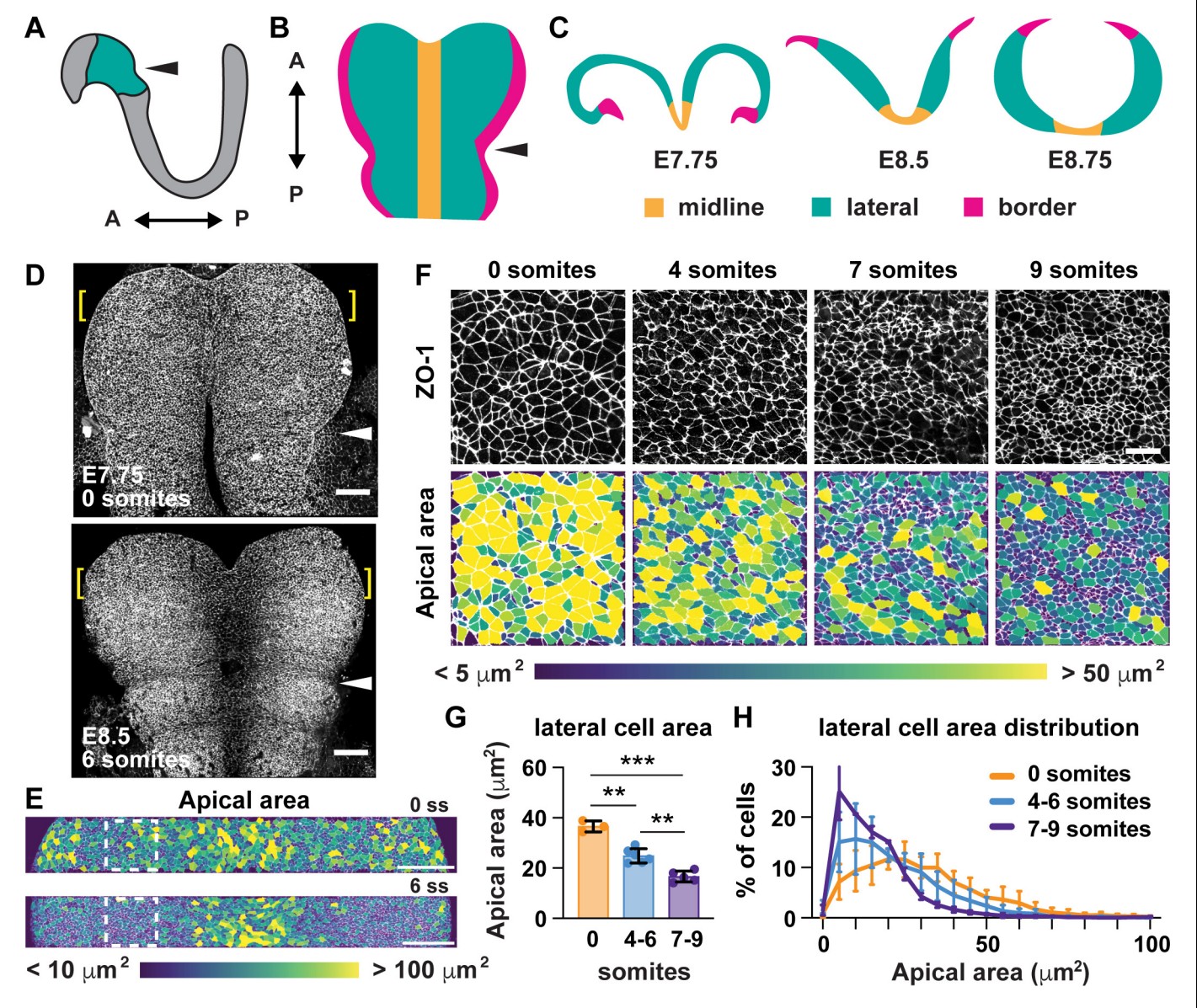

**Figure 1.** Lateral cells undergo apical remodeling during cranial neural fold elevation. (A) Schematic lateral view of the E8.5 neural plate showing the midbrain and anterior hindbrain region in green. (B) Schematic en face view of the midbrain and anterior hindbrain region. (C) Schematic cross-sectional views of the cranial neural plate during elevation. (D) Tiled confocal images of embryos at 0 somites (E7.75) and six somites (E8.5) labeled with ZO-1. Midline in center. Arrowhead, pre-otic sulcus. Brackets, regions shown in (E). (E) Midbrain cells color-coded by apical area. Boxes, regions shown in (F). (F) Lateral cells at progressive stages of neural fold elevation. Cells are labeled with ZO-1 (top) and are color-coded by apical area (bottom). (G,H) Average apical cell area (G) and apical area distributions (H) of lateral cells during midbrain neural fold elevation. A single value was obtained for each embryo and the mean ± SD between embryos is shown, n = 3–6 embryos/stage, **p<0.01, ***p<0.001 (one-way ANOVA test). See *Supplementary file 1* for n and p values. Anterior up in (D–F). Bars, 100 µm (D,E), 20 um (F).

used manual segmentation to investigate cell shape at the midline using antibodies to β-catenin (*Figure 3H*). In contrast to lateral cells, midline cells tended to be columnar (45 ± 5%) or apically expanded (30%), with relatively few midline cells showing apically constricted morphologies (17 ± 3%) (*Figure 3I*, *Figure 3—figure supplement 2A and B*). These results directly demonstrate the presence of apical constriction in the lateral neural plate and reveal a striking regionalization of cell-shape changes along the mediolateral axis.

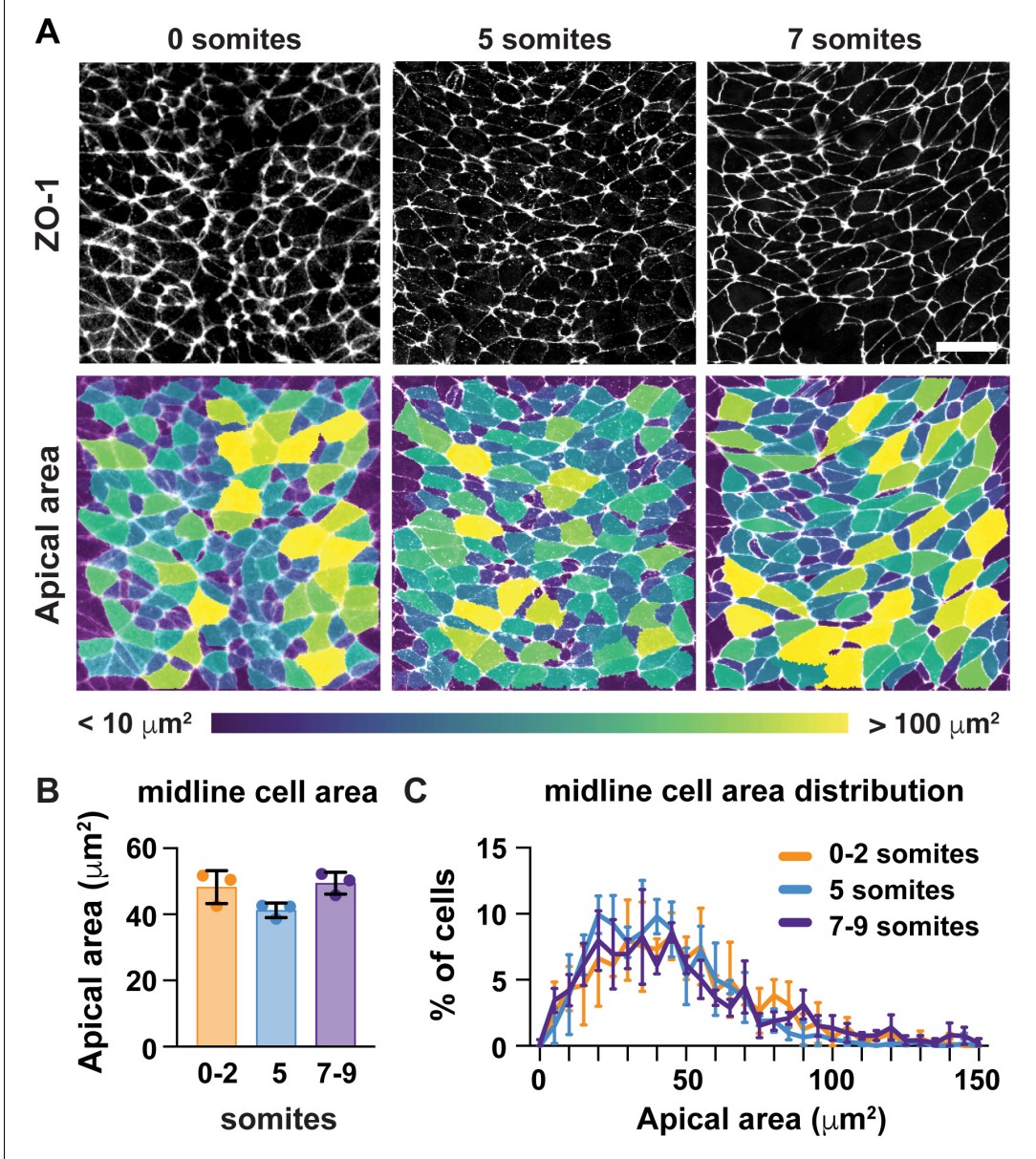

**Figure 2.** Midline cells do not undergo apical remodeling during cranial neural fold elevation. (A) Midline cells at progressive stages of neural fold elevation. Cells are labeled with ZO-1 (top) and are color-coded by apical area (bottom). (B,C) Average apical cell area (B) and apical area distributions (C) of midline cells during midbrain neural fold elevation. A single value was obtained for each embryo and the mean ± SD between embryos is shown, n = 3 embryos/stage, no significant differences (one-way ANOVA test). See *Supplementary file 1* for n and p values. Anterior up. Bar, 20 μm.

The online version of this article includes the following figure supplement(s) for figure 2:

**Figure supplement 1.** Analysis of mediolateral cell orientation in midline and lateral cells.

### Apical remodeling and cranial neural closure require IFT-A proteins

To identify the mechanisms that regulate the distinct behaviors of lateral and midline cells, we sought to identify mutants that disrupt this pattern. In a genetic screen for mouse mutants with embryonic defects (*García-García et al., 2005*), we identified two mutants with severe defects in cranial neural closure (*Figure 4A–C*). The mutations in these strains mapped to premature stop codons in *Ift122* and *Ttc21b (Ift139)*, which encode components of the conserved intraflagellar transport A (IFT-A) complex (*Figure 4—figure supplement 1A and B*). The IFT-A complex directs the trafficking of structural and signaling proteins in cilia, microtubule-based cellular organelles that modulate Shh

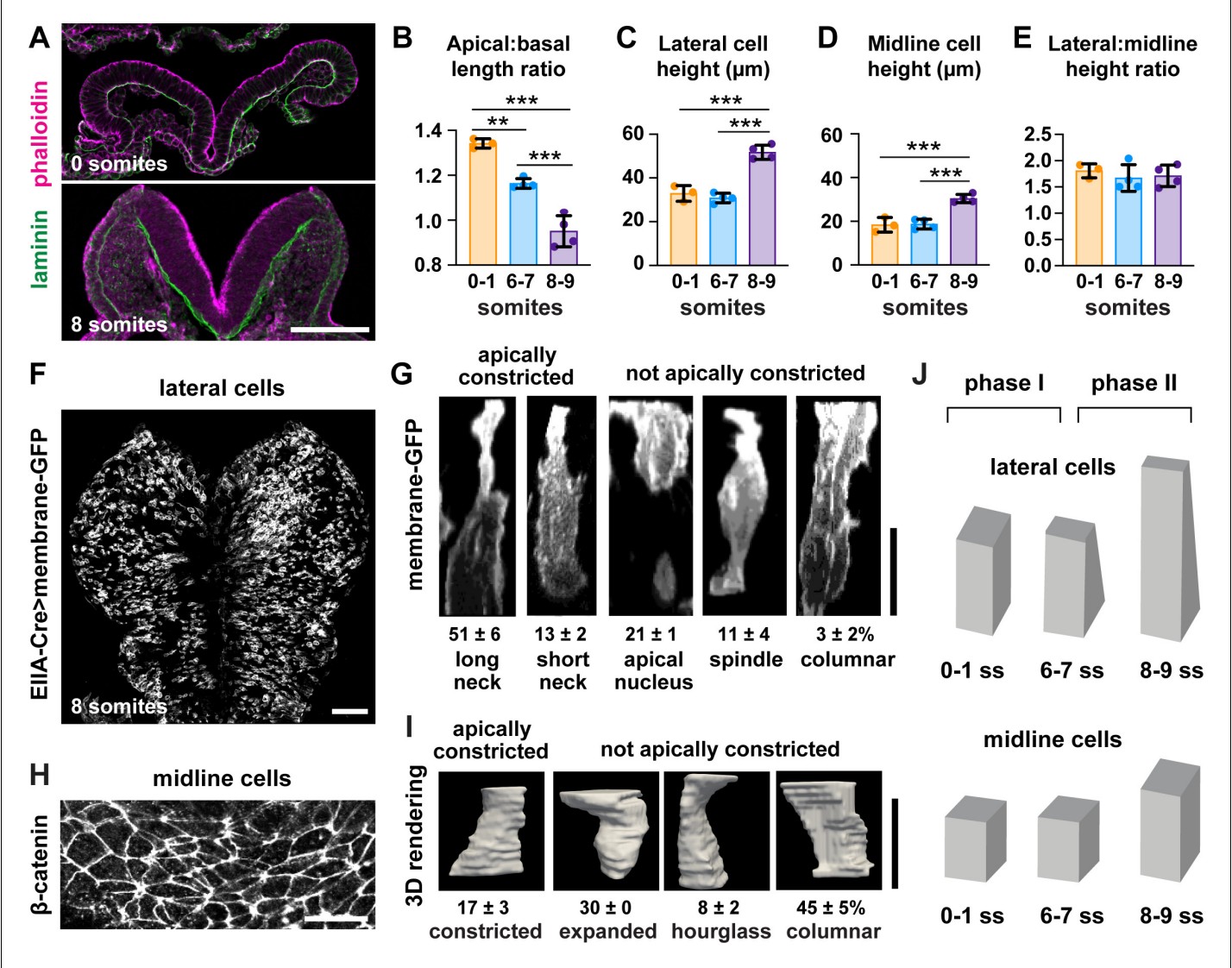

**Figure 3.** Lateral cells, but not midline cells, apically constrict. (A) Transverse sections of the cranial neural plate. Phalloidin and laminin show the apical and basal surfaces of the neuroepithelium, respectively. (B) The ratio of the apical span to the basal span of the neural plate decreases during elevation, flipping the cranial neural plate from convex (>1) to concave (<1). (C–E) Cell height in lateral (C) and midline (D) regions increases after the seven somite stage, but the ratio (E) does not change. (F) Mosaic expression of membrane-GFP using the EIIA-Cre driver. (G) 3D projections of membrane-GFP signal from individual lateral cells. (H) Midline cells labeled with β-catenin. (I) 3D surface renderings of manually segmented midline cells. (J) Midbrain neural fold elevation occurs in two phases. Early elevation (0–6 somites) is driven by apical constriction in lateral cells without a change in cell height. At later stages (7–9 somites), both midline and lateral cells undergo significant apicobasal cell elongation. A single value was obtained for each embryo and the mean ± SD between embryos is shown, n = 3–4 embryos/stage in (B–E), 408 cells in three embryos in (G), 60 cells in three embryos in (I), **p<0.01, ***p<0.001 (one-way ANOVA test). See *Supplementary file 1* for n and p values. Apical up in (A), (G), and (I), anterior up in (F) and (H). Bars, 100 μm (A,F), 20 μm (G–I).

The online version of this article includes the following figure supplement(s) for figure 3:

**Figure supplement 1.** The cranial neural plate transitions from convex to concave during elevation.

**Figure supplement 2.** Midline cells do not apically constrict during elevation.

signaling (*Wong and Reiter, 2008*; *Bangs and Anderson, 2017*). Consistent with these functions, mutant embryos from both strains exhibited a reduction in the number of cilia (*Figure 4—figure supplement 1C–E*). Mutations in IFT-A complex components have been shown to cause exencephaly, but how these proteins influence cranial neural closure is not known (*Tran et al., 2008*; *Cortellino et al., 2009*; *Murdoch and Copp, 2010*; *Qin et al., 2011*; *Bangs and Anderson, 2017*).

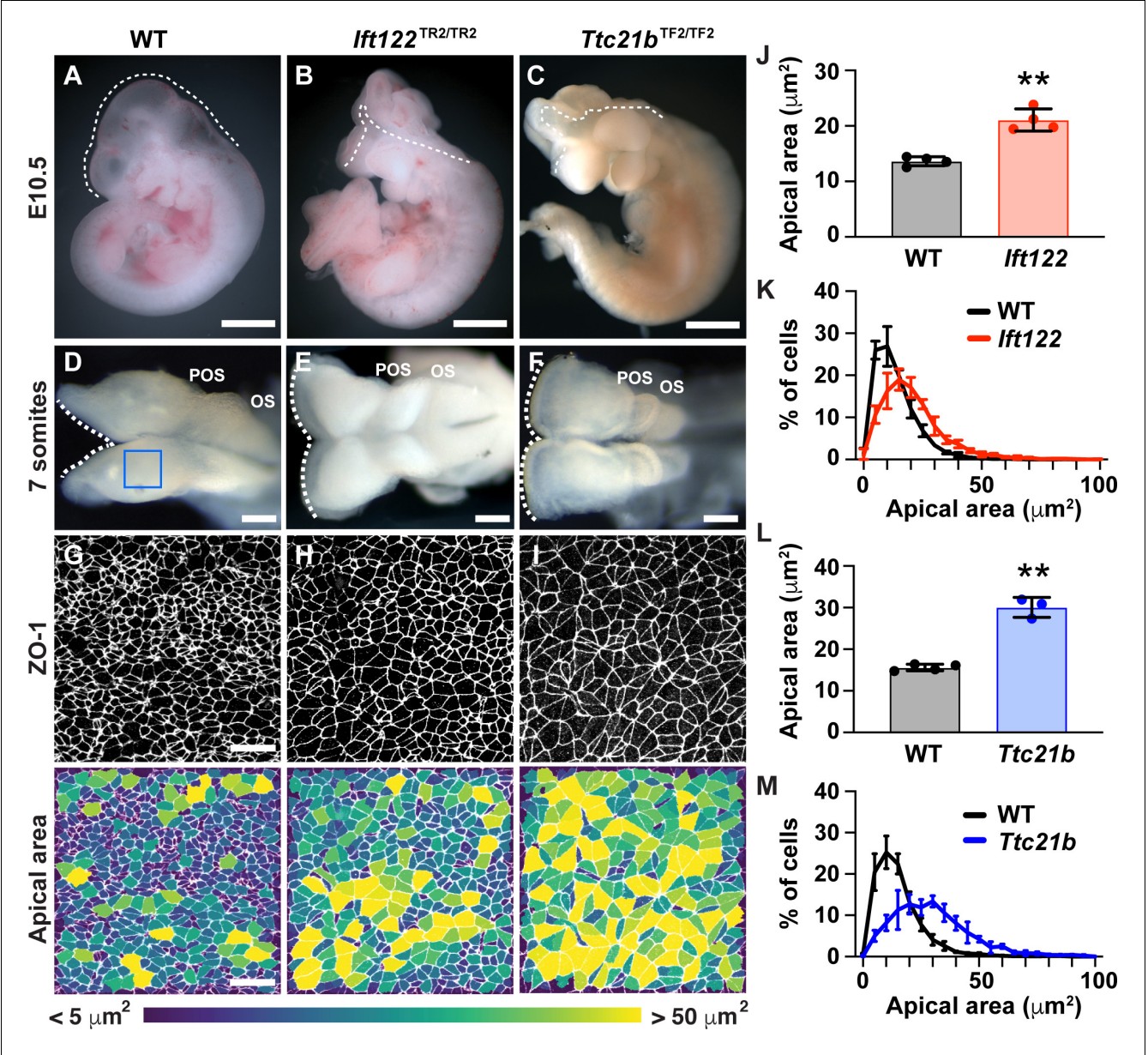

**Figure 4.** IFT-A proteins have an early role in cranial neural tube closure. (A) Wild-type littermate control (WT) showing normal cranial closure. (B,C) Exencephaly was observed in 10/10 *Ift122* mutants (B) (compared with 0/16 WT controls) and 5/5 *Ttc21b* mutants (C) (compared with 0/13 WT controls). Dashed lines, lateral edge of the cranial neuroepithelium. (D–F) The cranial neural folds fail to elevate in *Ift122* (E) and *Ttc21b* (F) mutants compared to WT controls (D). Box, region shown in (G–I). (G–I) Lateral cells in WT and mutant embryos. Cells are labeled with ZO-1 (top) and are color-coded by apical area (bottom). (J–M) Average apical cell area (J,L) and apical area distributions (K,M) of lateral cells in *Ift122* and *Ttc21b* mutants compared with WT controls. A single value was obtained for each embryo and the mean ± SD between embryos is shown, n = 3–4 embryos/genotype, **p<0.01 (Welch's t-test). See *Supplementary file 1* for n and p values. Anterior up in (A–C) and (G–I), anterior left in (D–F). Bars, 1 mm (A–C), 100 µm (D–F), and 20 µm (G–I).

The online version of this article includes the following figure supplement(s) for figure 4:

**Figure supplement 1.** Novel *Ift122* and *Ttc21b* alleles have defects in ciliogenesis.

**Figure supplement 2.** *Ift122* and *Ttc21b* mutants display a persistent failure of neural fold elevation.

**Figure supplement 3.** Disrupted cranial architecture in *Ift122* mutants.

**Figure supplement 4.** Analysis of mediolateral cell orientation in *Ift122* and *Ttc21b* mutants.

**Figure supplement 5.** Cell proliferation is not affected in *Ift122* and *Ttc21b* mutants.

**Figure supplement 6.** Mesenchymal cell density is not affected in *Ttc21b* mutants.

In contrast to littermate controls, which completed neural fold elevation, apposition, and fusion in 24 hr, *Ift122* and *Ttc21b* mutants failed to generate V-shaped neural folds in the midbrain, forebrain, and anterior hindbrain regions of 7-somite embryos (*Figure 4D–F*, *Figure 4—figure supplements 2* and *3*). These defects did not recover and the cranial neural folds of mutant embryos remained unelevated at all stages analyzed, leading to highly penetrant exencephaly at E10.5 (*Figure 4A–C*). Thus, the cranial closure defects in *Ift122* and *Ttc21b* mutants arise from an early failure in cranial neural fold elevation.

To determine the cellular basis of these exencephaly defects, we analyzed cell shape in *Ift122* and *Ttc21b* mutants. Mutant embryos displayed a striking expansion of the apical cell surface in the lateral midbrain (*Figure 4G–I*). Lateral cells in *Ift122* and *Ttc21b* mutants displayed a 55% and 93% increase in average apical cell area, respectively, compared with littermate controls (*Figure 4J–M*), as well as altered cell orientation (*Figure 4—figure supplement 4*). These defects did not result from reduced cell proliferation, as mutant embryos had a normal frequency and distribution of mitotic cells along the mediolateral axis, and normal cell density in the underlying mesenchyme (*Figure 4—figure supplements 5* and *6*). These results indicate that *Ift122* and *Ttc21b* are required for cell-shape changes in the lateral midbrain neuroepithelium.

## IFT-A proteins pattern cell shape and actomyosin contractility

To determine if the global pattern of cell remodeling is affected in IFT-A mutants, we examined cell-shape changes throughout the entire mediolateral axis of the midbrain in *Ift122* and *Ttc21b* mutants. In wild-type littermate controls, apically expanded cells were present at the midline and at the outer margins of the tissue. These domains were separated by a broad domain of apically constricted cells spanning 30–40 cell diameters along the mediolateral axis and more than 60 cells along the anterior-posterior axis, encompassing a region of more than 2000 lateral cells on either side of the midline (*Figure 5A and F–H*). The difference between midline and lateral populations was eliminated in *Ift122* and *Ttc21b* mutants (*Figure 5B–E*). In mutant embryos, lateral cells were apically expanded and midline cells were apically constricted compared with controls, whereas cell shape at the outer margins of the neural plate was independent of IFT-A activity (*Figure 5F–H*). Moreover, the difference in height between wild-type midline and lateral cells was abolished in *Ift122* and *Ttc21b* mutants (*Figure 5I and J*, *Figure 5—figure supplement 1A and B*). These cell remodeling defects were associated with a failure to fully convert the cranial region from convex to concave in *Ift122* and *Ttc21b* mutants (*Figure 5—figure supplement 1C–F*). These results demonstrate that Ift122 and Ttc21b are required for patterned apical remodeling in the midbrain neuroepithelium. In their absence, midline and lateral cells adopt a uniform cell morphology.

A hallmark of apical constriction is the requirement for apically localized actomyosin contractility (*Martin and Goldstein, 2014*). To determine if this is the mechanism by which Ift122 and Ttc21b promote apical remodeling in lateral cells, we analyzed the localization of F-actin and the phosphory-lated (active) form of myosin II in *Ift122* and *Ttc21b* mutants. Wild-type cranial neuroepithelial cells display a strong accumulation of F-actin and phosphomyosin at the apical cell cortex, which is often assembled into mediolaterally oriented actomyosin cables in the chick and mouse neural plate (*Nishimura et al., 2012*; *McGreevy et al., 2015*). In line with these observations, we observed frequent supracellular cables in the elevating cranial neural plate. Cables were present at a range of orientations, with a strong mediolateral bias (*Figure 6A,B,E and F*). By contrast, fewer actomyosin cables were present in *Ift122* and *Ttc21b* mutants, and the cables that did form were not consistently oriented with respect to the mediolateral axis (*Figure 6A–F*). The ratio of phosphomyosin to F-actin at adherens junctions was also decreased in *Ttc21b* mutants (*Figure 6G and H*), consistent with the stronger apical constriction defects in this mutant. These results demonstrate that Ift122 and Ttc21b are required for apical actomyosin organization in the midbrain neuroepithelium.

## Shh signaling organizes patterned apical remodeling

Cilia are signaling organelles that are critical for Shh signaling and cell fate. Our finding that cilia proteins are also required for cell remodeling suggests that fate and morphology may be directly linked. In the spinal cord, the loss of IFT-A complex function typically results in ligand-independent activation of Shh signaling and an expansion of Shh-dependent ventral cell fates (*Tran et al., 2008*; *Cortellino et al., 2009*; *Qin et al., 2011*; *Bangs and Anderson, 2017*), although strong disruption

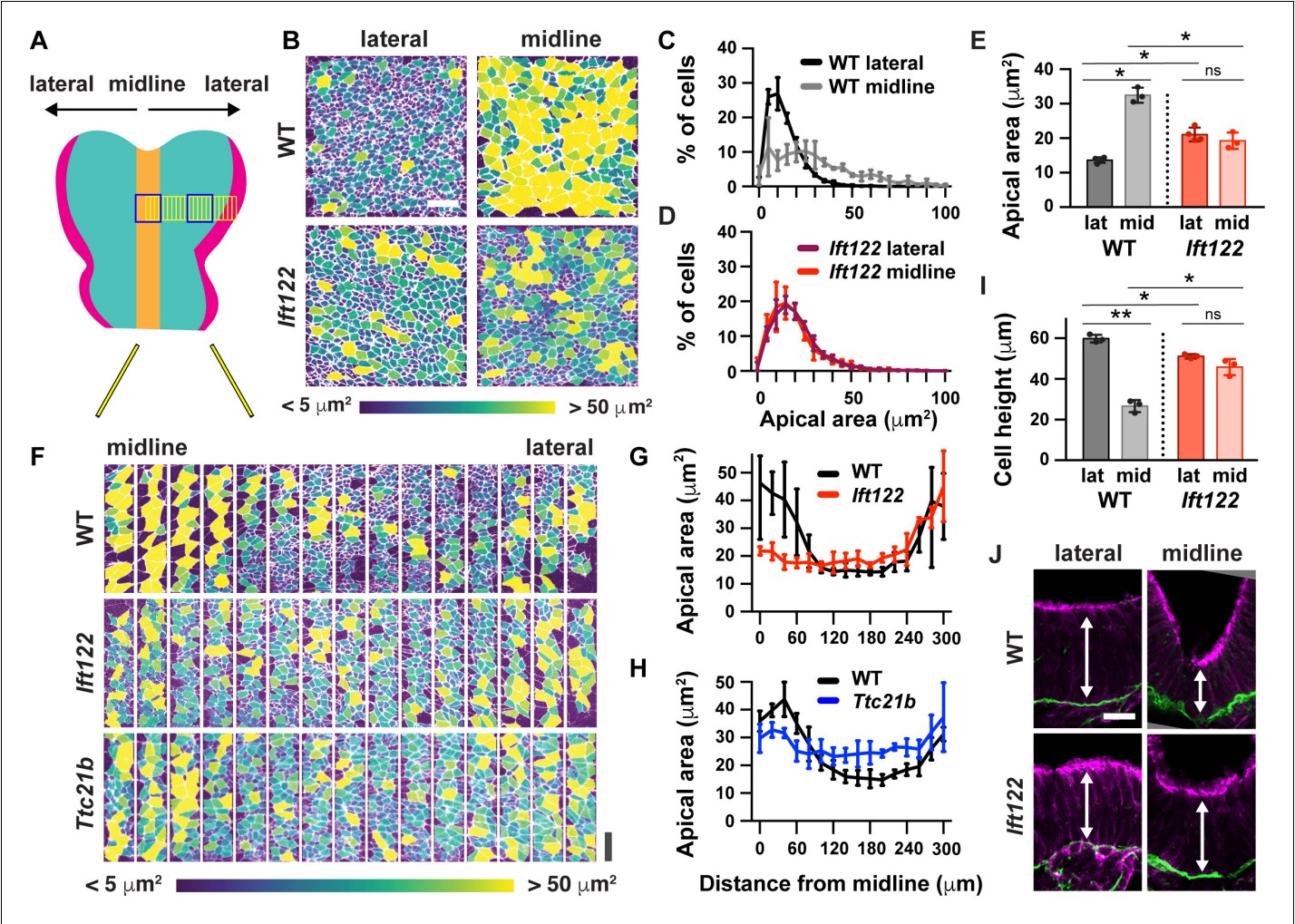

**Figure 5.** IFT-A mutants display a failure of patterned apical remodeling. (**A**) Schematic of midbrain regions analyzed in (**B**) (blue boxes) and (**F**) (yellow boxes). (**B**) Lateral and midline cells labeled with ZO1 are color-coded by apical area in wild-type littermate control (WT) and *Ift122* mutant embryos. (**C**–**E**) Apical area distributions (**C,D**) and average apical cell area (**E**). Lateral measurements are reproduced from *Figure 4J*. (**F**) Contiguous 20 µm wide regions spanning the mediolateral axis from the midline to the lateral margins of the midbrain neural plate. Cells are labeled with ZO1 and color-coded by apical area. (**G,H**) Apical cell area plotted by distance from the midline. (**I,J**) Average cell height in midline and lateral cells (**I**) measured in transverse sections of the cranial neural plate (**J**). Phalloidin and laminin show apical and basal surfaces, respectively. A single value was obtained for each embryo and the mean ± SD between embryos is shown, n = 3–4 embryos/genotype, *p<0.05, **p<0.01 (one-way ANOVA test). See *Supplementary file 1* for n and p values. Embryos are anterior up, 7 somites (**B–H**) or apical up, 12 somites (**I,J**). Bars, 20 µm.

The online version of this article includes the following figure supplement(s) for figure 5:

**Figure supplement 1.** The convex to concave transition is defective in *Ift122* and *Ttc21b* mutants.

of IFT-A function can result in a loss of Shh-dependent cell fates (*Liem et al., 2012*). To test whether Ift122 and Ttc21b pattern cell fate during cranial neural fold elevation, we analyzed the expression of Nkx6.1, a target of Shh signaling. In wild-type embryos, Nkx6.1 levels were highest at the midline during midbrain neural fold elevation and decreased with increasing distance from the midline (*Figure 7A and B*), consistent with results at later stages (*Qiu et al., 1998*; *Tran et al., 2008*; *Qin et al., 2011*; *Tang et al., 2013*). In addition, Nkx6.1 was expressed at lower levels in the anterior hindbrain, revealing differential regulation along the anterior-posterior axis (*Figure 7A*). In *Ift122* and *Ttc21b* mutants, the mediolateral pattern of Nkx6.1 expression was abolished, and Nkx6.1 was expressed at equivalent, intermediate levels in midline and lateral cells (*Figure 7A–C*). In addition, the mediolateral extent of the Nkx6.1 domain was expanded, reaching all the way to the neural plate borders in *Ift122* and *Ttc21b* mutants. By contrast, the anterior-posterior pattern of Nkx6.1

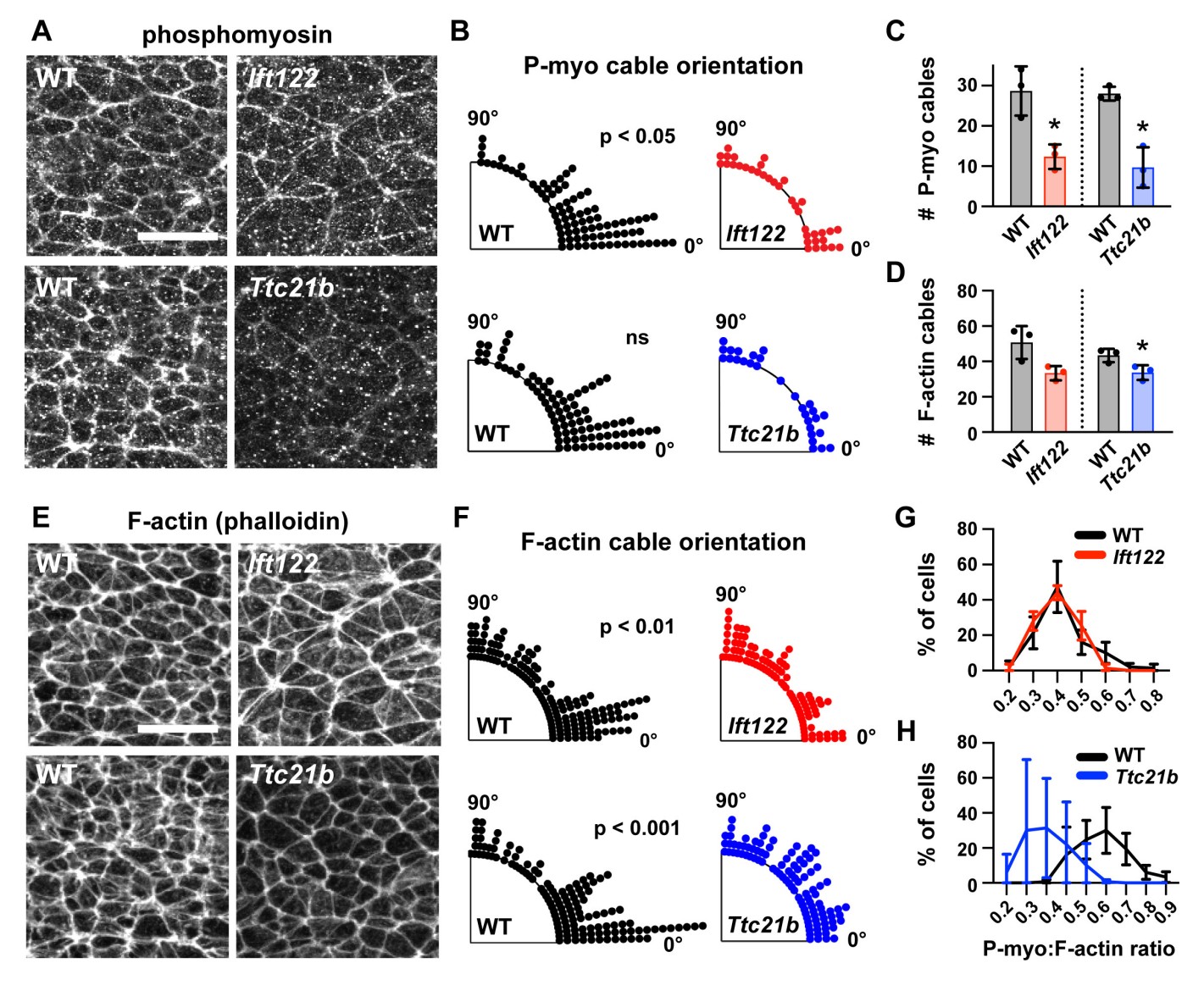

**Figure 6.** Actomyosin organization is disrupted in IFT-A mutants. (A,E) Localization of phosphorylated myosin II (phosphomyosin) (A) and F-actin (E) in lateral cells of *Ift122* and *Ttc21b* mutants and wild-type littermate controls (WT). (B,F) Orientation of apical phosphomyosin (P-myo) cables (B) and F-actin cables (F) in *Ift122* and *Ttc21b* mutants and WT controls. (C,D) The number of phosphomyosin cables (C) and F-actin cables (D) per embryo in two 100 μm x 100 μm lateral regions in *Ift122* and *Ttc21b* mutants. (G,H) The ratio of phosphomyosin to F-actin at cell-cell junctions was shifted to lower values in *Ttc21b* mutants. A single value was obtained from each embryo and the mean ± SD between embryos is shown, n = 29–86 phosphomyosin cables and 100–151 F-actin cables from three embryos/genotype in (A–F), 50 cells from three embryos/genotype (G,H), *p<0.05, Welch's t-test in (C,D), Watson two-sample test for homogeneity (B,F). See *Supplementary file 1* for n and p values. Embryos are 7-8 somites. Anterior up. Bars, 20 μm.

expression was unaffected, indicating that this axis of Shh regulation is independent of IFT-A activity. These results raise the possibility that deregulated Shh signaling could underlie the cranial closure defects in *Ift122* and *Ttc21b* mutants.

To investigate the role of Shh signaling in midbrain cell remodeling, we examined cell morphology in mutants lacking the Shh effector Gli2, which is required to generate ventral Shh-dependent cell types (*Mo et al., 1997*; *Matise et al., 1998*; *Bai and Joyner, 2001*; *Wong and Reiter, 2008*; *Bangs and Anderson, 2017*). Because Shh signaling is normally highest at the midline, we asked if the unique architecture of midline cells requires Gli2 function. Consistent with the effects of Gli2 at later stages, *Gli2* mutants failed to establish ventral cell fates in the elevating midbrain, including FoxA2 expression in the floor plate, indicating that Gli2 is required for midline cell identity

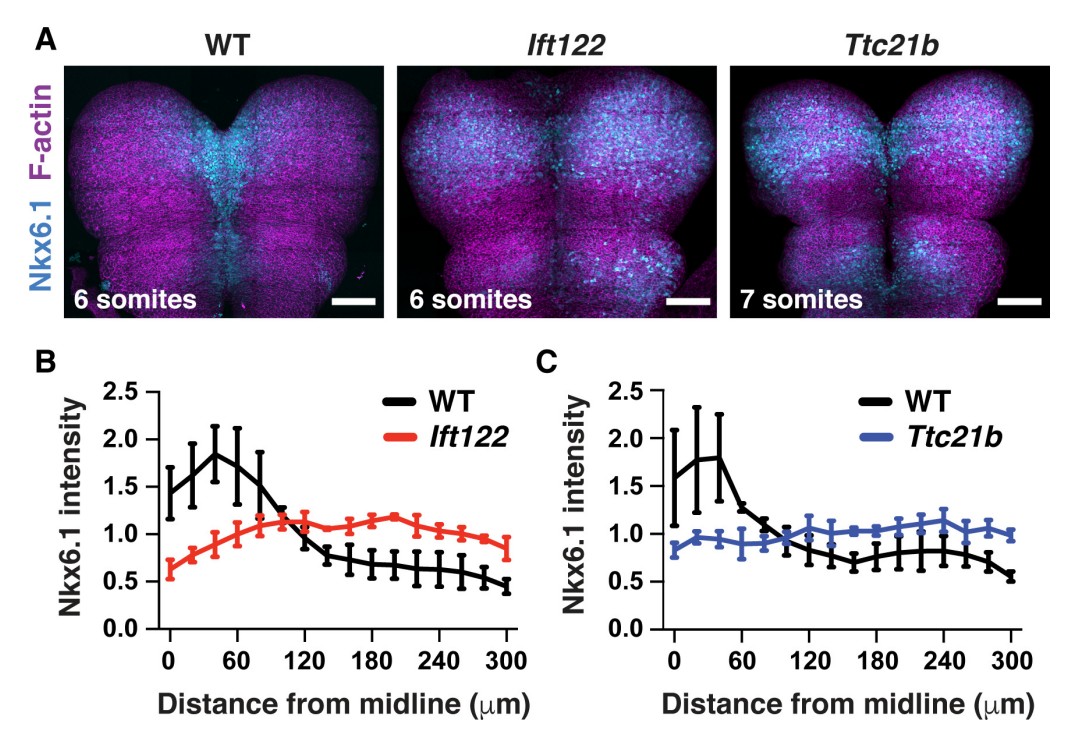

**Figure 7.** Shh-dependent cell fates expand laterally in *Ift122* and *Ttc21b* mutants. (A) Nkx6.1 protein visualized in tiled confocal images of cranial neural plate cells labeled with phalloidin (F-actin). (B,C) Nkx6.1 intensity plotted by distance from the midline, normalized to the mean Nkx6.1 intensity of the image, in *Ift122* (B) and *Ttc21b* (C) mutants compared with wild-type littermate controls (WT). A single value was obtained for each embryo and the mean ± SD between embryos is shown, n = 3 embryos/genotype. Anterior up. Bars, 100 μm.

(*Figure 8—figure supplement 1*). Midline cells in *Gli2* mutants had a significant decrease in apical area compared with wild type (*Figure 8D–F*). In addition, midline cells in *Gli2* mutants were significantly taller than in wild-type controls (*Figure 8G and H*). By contrast, lateral cell morphology was unaffected in *Gli2* mutants (*Figure 8A–C,G and H*), and no defects in cell orientation were observed in either region (*Figure 8—figure supplement 2*). These data demonstrate that Gli2 activity is necessary for the short, apically expanded architecture of midline cells but not for apical constriction in lateral cells. Despite these severe midline defects, *Gli2* mutants complete closure normally in both cranial and spinal regions (*Mo et al., 1997*; *Matise et al., 1998*; *Bai and Joyner, 2001*), indicating that the specialized architecture of midline cells is dispensable for cranial neural tube closure.

Because cranial neural tube closure occurs normally in the absence of proper midline morphology, we hypothesized that a failure of apical constriction in lateral cells could be responsible for the cranial closure defects in *Ift122* and *Ttc21b* mutants. In addition, the expanded Nkx6.1 expression in *Ift122* and *Ttc21b* mutants raises the possibility that increased Shh signaling in lateral cells could underlie the defects in apical constriction. To test these hypotheses, we investigated whether spatially restricted Shh signaling is required for apical constriction and cranial neural closure. We ectopically activated the Shh signaling response throughout the midbrain by expressing a constitutively active variant of the Shh receptor Smoothened (SmoM2) (*Jeong et al., 2004*) using Wnt1-Cre2 (*Lewis et al., 2013*). SmoM2-expressing embryos have expanded Nkx6.1 expression throughout the midbrain, consistent with uniform activation of the Shh response (*Figure 9C and D*). SmoM2 expression did not affect cell proliferation or cell orientation (*Figure 9—figure supplements 1* and *2*). However, SmoM2 expression resulted in a 50% increase in apical area in lateral cells (*Figure 9G–I*), similar to the defects in *Ift122* mutants, but less severe than the defects in *Ttc21b* mutants. The effects of SmoM2 were localized, as cells that did not express Wnt1-Cre2 apically constricted normally, suggesting that activated SmoM2 acts cell autonomously to regulate cell shape (*Figure 9E and F*). SmoM2-expressing embryos did not display morphological defects at the midline, perhaps because Smoothened activation did not further enhance the already high Shh response in this region

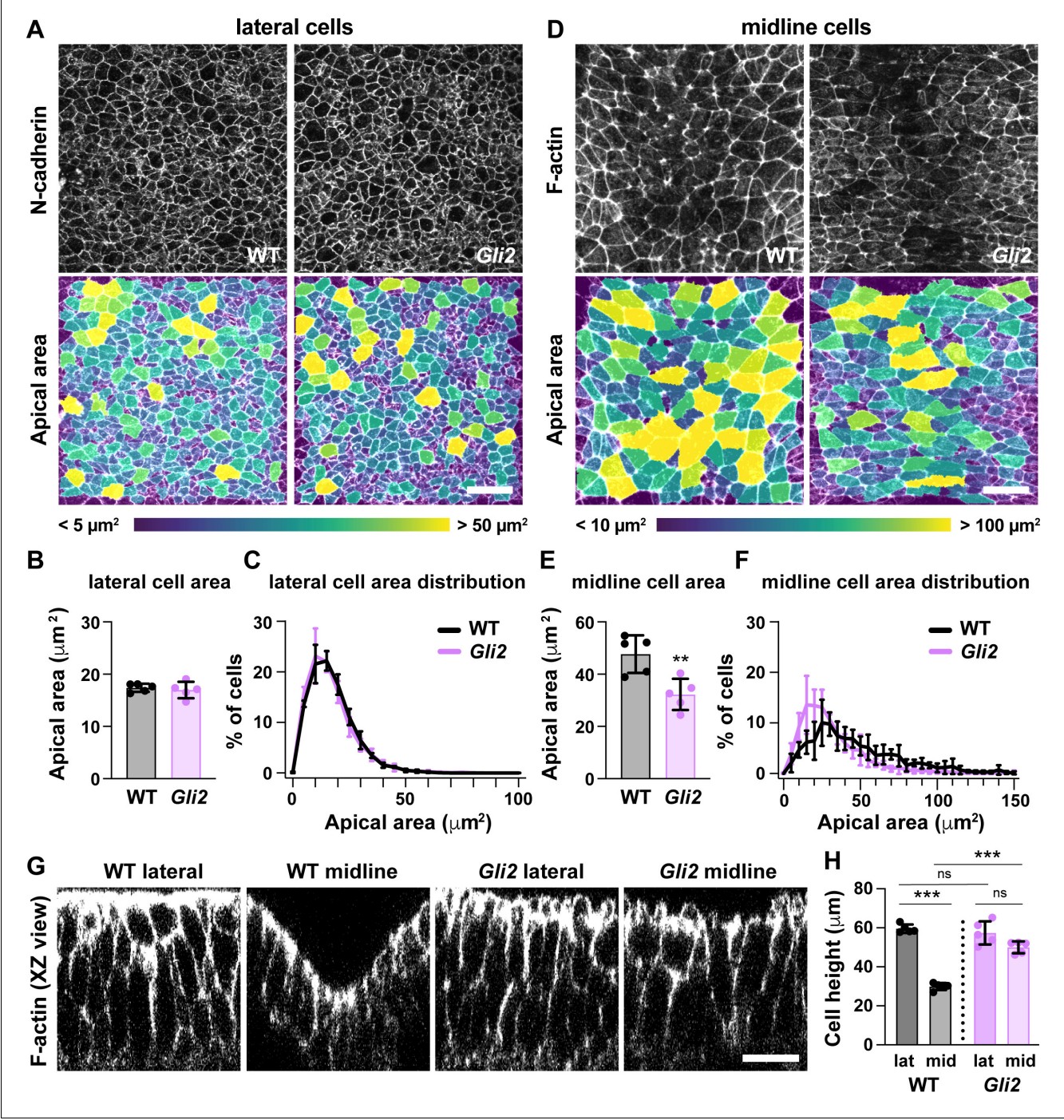

**Figure 8.** Loss of Gli2 disrupts midline but not lateral cell shape. (A,D) Lateral (A) and midline (D) cells in *Gli2* mutant embryos and wild-type littermate controls (WT). Cells are labeled with N-cadherin or phalloidin (top) and color coded by apical area (bottom). (B,C,E,F) Average apical cell area (B,E) and apical area distributions (C,F) of lateral and midline cells from WT and *Gli2* mutant embryos. (G) XZ reconstructions of lateral and midline cells labeled with F-actin in WT and *Gli2* mutant embryos. (H) Cell height in lateral and midline cells in WT and *Gli2* mutant embryos. A single value was obtained for each embryo and the mean ± SD is shown. n = 5 embryos/genotype, **p<0.01, ***p<0.001, Welch's t-test (B,E) or Brown-Forsythe one-way ANOVA test (H). See *Supplementary file 1* for n and p values. Embryos are 7–9 somites. Anterior up in (A,D), apical up in (G). Bars, 20 µm.

The online version of this article includes the following figure supplement(s) for figure 8:

**Figure supplement 1.** Gli2 is required for FoxA2 expression in the ventral neural plate.

**Figure supplement 2.** Analysis of mediolateral cell orientation in *Gli2* mutants.

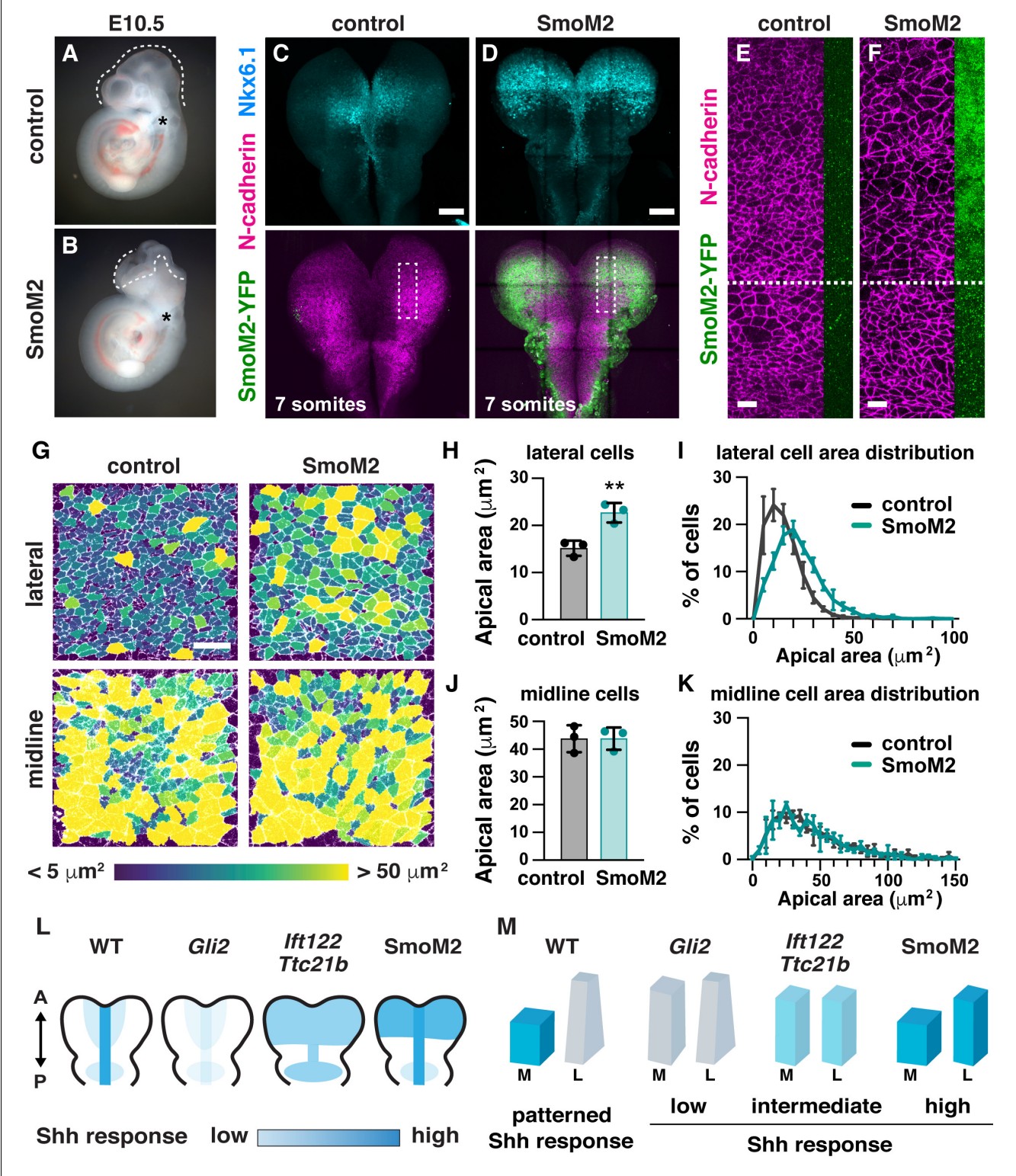

**Figure 9.** Ectopic Shh signaling disrupts lateral cell remodeling and causes exencephaly. (**A,B**) Expression of the activated Shh receptor Smoothened (SmoM2) using the midbrain-specific Wnt1-Cre2 driver causes exencephaly (12/12 Wnt1-Cre2; SmoM2 embryos vs. 0/13 littermate controls). Control embryos were Wnt1-Cre2 or SmoM2 alone. (**C,D**) Wnt1-Cre2 drives SmoM2-YFP expression in the midbrain and induces ectopic Nkx6.1 expression throughout the mediolateral axis. Boxes, regions shown in (**E,F**). (**E,F**) Cells expressing SmoM2-YFP have larger apical areas compared with cells outside of the Wnt1-Cre2 expression domain (cells below the dashed line) and cells from equivalent regions in controls (**E**). SmoM2-YFP signal at the lateral

*Figure 9 continued on next page*

*Figure 9 continued*

edge of the N-cadherin region is shown. (**G**) Lateral and midline cells labeled with N-cadherin are color coded by area in control and SmoM2-expressing embryos. (**H–K**) Average apical cell area (**H,J**) and apical area distributions (**I,K**) in lateral and midline cells in control and SmoM2-expressing embryos. (**L**) Schematics of the pattern and intensity of the Shh response in WT, *Gli2* mutant, IFT-A mutant, and SmoM2-expressing embryos. (**M**) Model. The different shapes of lateral and midline cells correlate with different levels of Shh signaling. A high Shh response inhibits apical remodeling and apicobasal elongation in midline cells, whereas a low Shh response allows apical constriction in lateral cells. A single value was obtained for each embryo and the mean ± SD between embryos is shown, n = 3 embryos/genotype, \*\*p<0.01 (Welch's t-test). See *Supplementary file 1* for n and p values. Embryos are E10.5 in (**A,B**), 6–7 somites in (**E–K**). Anterior up. Bars, 100 μm in (**C,D**), and 20 μm in (**E–G**).

The online version of this article includes the following figure supplement(s) for figure 9:

**Figure supplement 1.** Cell proliferation is not affected by neuroepithelial expression of SmoM2.
**Figure supplement 2.** Analysis of mediolateral cell orientation in SmoM2-expressing embryos.

(*Figure 9G,J and K*). Notably, SmoM2-expressing embryos exhibited 100% penetrant exencephaly (12/12 SmoM2-expressing embryos compared with 0/13 littermate controls) (*Figure 9A and B*). These results demonstrate that disruption of apical constriction in lateral cells alone—in the absence of structural changes at the midline—is sufficient to prevent cranial neural closure. Thus, patterned Shh signaling in the midbrain neuroepithelium is required for spatially regulated apical remodeling events that drive cranial neural closure, and dysregulation of Shh activity leads to altered cell remodeling and exencephaly (*Figure 9L and M*).

## Discussion

Neural tube closure defects are among the most common human birth defects, with one-third of cases arising from defects in closure of the cranial region (*Zaganjor et al., 2016*). However, the mechanisms that convert the large cranial neural plate region from convex to concave during neural tube closure have long been obscure. Here, we show that elevation is driven by a tissue-scale pattern of apical cell remodeling in the mouse midbrain in which thousands of lateral cells undergo synchronous, sustained apical constriction, whereas midline cells remain apically expanded. Spatiotemporally regulated cell remodeling in this system requires patterned Shh signaling. Loss of the Shh effector Gli2 results in a failure to establish the short and flat morphology of cells at the midline, but does not prevent neural tube closure. By contrast, expansion of Shh signaling into the lateral neural plate, either in IFT-A mutants that impair cilia-dependent Shh regulation or in embryos that express activated Smoothened throughout the midbrain neuroepithelium, leads to a disruption of apical constriction in lateral cells and results in highly penetrant exencephaly. These results reveal a program of positionally encoded cell behavior that is essential for neural tube closure in the developing midbrain and identify Shh as a critical regulator of coordinated cell remodeling in the mammalian cranial neural plate.

Lateral cells that undergo apical constriction and midline cells that do not are distinguished by their distance from the source of the Shh signal, as the Shh response is normally high at the midline and diminished laterally (*Qiu et al., 1998*; *Sagner and Briscoe, 2019*; *Tang et al., 2013*). The morphological changes in embryos with a reduced or expanded response to Shh signaling are consistent with a model in which high levels of Shh signaling induce short, apically expanded cells, whereas low levels of Shh signaling are associated with tall, apically constricted cells. *Gli2* mutants that disrupt specific aspects of Shh signaling are defective for cell-shape changes in midline cells where Shh signaling is normally high (*Figure 9L and M*). By contrast, in SmoM2-expressing embryos that have aberrantly high Shh signaling throughout the midbrain, midline morphology is normal, but lateral cells fail to apically constrict. IFT-A mutants that have a uniform, intermediate level of Shh signaling throughout the midbrain display equivalent cell morphologies in both regions. These results are consistent with a model in which different levels of Shh signaling induce different cell shapes in the midbrain neuroepithelium, with high levels of Shh signaling inhibiting apical remodeling and apicobasal cell elongation at the midline, and low levels of Shh signaling allowing apical constriction in lateral cells. Thus, Shh not only determines the pattern of cell fates in the tissue, but is also essential for the organized cell behaviors that establish tissue structure. These dual functions of Shh provide a single source of positional information that regulates both cell identity and cell morphology, linking tissue pattern to tissue structure.

Apical constriction is a potent and conserved mechanism for generating changes in cell shape (*Martin and Goldstein, 2014*). During morphogenesis, apical constriction in a narrow or spatially delimited domain promotes localized tissue bending or invagination, as in *Drosophila* ventral furrow formation (*Ko and Martin, 2020*), *C. elegans* gastrulation (*Lee and Goldstein, 2003*; *Lee et al., 2006*), *Xenopus* blastopore invagination (*Keller, 1981*; *Lee and Harland, 2007*; *Lee and Harland, 2010*), and hinge point formation in the vertebrate spinal cord (*Haigo et al., 2003*; *Lee et al., 2007*; *Vijayraghavan and Davidson, 2017*). By contrast, we show that widespread apical constriction events are coordinated across thousands of cells in the developing midbrain, resulting in a large-scale change in the curvature of the elevating neural plate. Coordinated apical constriction in large cell populations has been observed in tissues that undergo a dramatic change or even an inversion of tissue curvature, such as in the mouse lens placode, which transitions from flat to spherical (*Plageman et al., 2010*), and in colonies of adherent unicellular choanoflagellates undergoing light-dependent curvature inversion (*Brunet et al., 2019*). Thus, coordinated constriction among hundreds to thousands of cells may represent an evolutionarily conserved mechanism for collectively promoting large-scale curvature changes in multicellular tissues.

The cell-shape defects caused by an expanded Shh response in IFT-A mutants suggest a unifying hypothesis for the cranial closure defects in mutants with deregulated Shh signaling (*Murdoch and Copp, 2010*), including mutants that affect cilia structure (*Liem et al., 2012*), transducers of the Shh signal such as Gli3 and Sufu (*Hui and Joyner, 1993*; *Svärd et al., 2006*), and negative regulators of the Shh response (*Ikeda et al., 2001*; *Cameron et al., 2009*; *Norman et al., 2009*; *Patterson et al., 2009*). Expanded Shh signaling could inhibit apical constriction through a canonical signaling pathway involving Gli2- and Gli3-mediated transcriptional changes (*Dessaud et al., 2008*; *Kicheva and Briscoe, 2015*; *Bangs and Anderson, 2017*; *Sagner and Briscoe, 2019*), possibly involving repression of the BMP inhibitor Noggin, which promotes tissue bending in the spinal cord (*Ybot-Gonzalez et al., 2002*; *Ybot-Gonzalez et al., 2007*; *Eom et al., 2011*). Consistent with this possibility, the loss of Noggin has been shown to cause exencephaly (*Stottmann et al., 2006*). Alternatively, Shh could regulate cell shape through a noncanonical signaling pathway (*Robbins et al., 2012*; *de la Roche et al., 2013*; *Zuñiga and Stoeckli, 2017*). Elucidation of the effector pathways that generate cell shape downstream of Shh signaling will reveal how cell morphology and cell fate are coordinately regulated in response to the Shh signal.

Shh signaling has long been recognized to play an important role in controlling positional cell fates in many developing organs, including the limb, the gut, and the spinal cord (*Jessell, 2000*; *Villavicencio et al., 2000*; *McMahon et al., 2003*; *McGlinn and Tabin, 2006*; *Tickle and Towers, 2017*; *Sagner and Briscoe, 2019*). Although the effects of Shh on cell behavior have received comparatively less attention, Shh has been shown to influence axon guidance (*Zuñiga and Stoeckli, 2017*), cell migration (*Gordon et al., 2018*), mesenchymal cell clustering (*Rao-Bhatia et al., 2020*), and epithelial remodeling in mice, chicks, frogs, and flies (*Corrigall et al., 2007*; *Escudero et al., 2007*; *Nasr et al., 2019*; *Arraf et al., 2020*). Depending on the context, proteins in the Shh family can have contrasting effects on epithelial cell behavior, promoting apical constriction in the *Drosophila* eye (*Corrigall et al., 2007*; *Escudero et al., 2007*), generating short and flat cells in the neural tube (*Fournier-Thibault et al., 2009* and this work), and inducing tall, pseudostratified cells in the chick coelomic cavity (*Arraf et al., 2020*). An understanding of the mechanisms by which Shh signaling directs cell morphology will provide insight into how this conserved, positionally encoded molecular mechanism coordinates cell fate with three-dimensional tissue structure.

## Materials and methods

### Key resources table

| Reagent type (species) or resource | Designation | Source or reference | Identifiers | Additional information |
|---|---|---|---|---|
| Genetic reagent (*Mus musculus*) | FVB/NJ | Jackson Laboratory | stock no. 001800 RRID:IMSR_JAX:001800 | |
| Genetic reagent (*Mus musculus*) | *Iftt122^{TR2}* | This study | | FVB/N background |

*Continued on next page*

*Continued*

| Reagent type (species) or resource | Designation | Source or reference | Identifiers | Additional information |
|---|---|---|---|---|
| Genetic reagent (*Mus musculus*) | *Ttc21b^TF2* | This study | | FVB/N background |
| Genetic reagent (*Mus musculus*) | SmoM2 | Jackson Laboratory (*Jeong et al., 2004*) | *Gt(ROSA) 26Sor^tm1(Smo/EYFP)Amc*/J stock no. 005130 MGI:3576373 RRID:IMSR_JAX:005130 | C57BL/6J background |
| Genetic reagent (*Mus musculus*) | *Wnt1-Cre2* | Jackson Laboratory (*Lewis et al., 2013*) | *E2f1^Tg(Wnt1-cre)2Sor*/J stock no. 022137 MGI:5485027 RRID:IMSR_JAX:022137 | FVB/N background |
| Genetic reagent (*Mus musculus*) | EIIA-Cre | Jackson Laboratory (*Lakso et al., 1996*) | Tg(EIIa-cre) C5379Lmgd/J stock no. 003314 MGI:2137691 RRID:IMSR_JAX:003314 | FVB/N background |
| Genetic reagent (*Mus musculus*) | *mT/mG* | Jackson Laboratory (*Muzumdar et al., 2007*) | *Gt(ROSA) 26Sor^tm4(ACTB-tdTomato,-EGFP)Luo*/J stock no. 007676 MGI:3716464 RRID:IMSR_JAX:007676 | FVB/N background |
| Genetic reagent (*Mus musculus*) | *Gli2^lzki* | Jackson Laboratory (*Bai and Joyner, 2001*) | Gli2^tm2.1Alj/J stock no. 007922 MGI:3815004 RRID:IMSR_JAX:007922 | SWR/J background |
| Antibody | Anti-ZO-1 (rat monoclonal) | Developmental Studies Hybridoma Bank (DSHB) | R26.4C RRID:AB_2205518 | (1:100) |
| Antibody | Anti-phospho-Histone H3 (rabbit polyclonal) | Upstate | 06–570 RRID:AB_310177 | (1:1000) |
| Antibody | Anti-Arl13b (rabbit polyclonal) | *Caspary et al., 2007* | | (1:1000) |
| Antibody | Anti-β-catenin (mouse monoclonal) | BD | 610153 RRID:AB_397554 | (1:300) |
| Antibody | Anti-laminin (rabbit polyclonal) | Sigma | L9393 RRID:AB_477163 | (1:1000) |
| Antibody | Anti-N-cadherin (rabbit monoclonal) | Cell Signaling Technology | D4R1H RRID:AB_2687616 | (1:500) |
| Antibody | Anti-GFP (chicken polyclonal) | abcam | ab13970 RRID:AB_300798 | (1:1000) |
| Antibody | Anti-Nkx6.1 (mouse monoclonal) | DSHB | F55A10 RRID:AB_532378 | (1:50) |
| Antibody | Anti-diphospho myosin regulatory light chain (rabbit polyclonal) | Cell Signaling Technology | 3674 RRID:AB_2147464 | (1:100) |
| Antibody | Anti-FoxA2 (rabbit monoclonal) | abcam | ab108422 RRID:AB_11157157 | (1:1000) |
| Sequence-based reagent | Ift122(TR2)_F | This study | PCR primer | CTGGTTGTAAT CTGACTCGTTGA After amplification with below reverse primer, product is digested with HpyCH4III, resulting in a 133 bp WT band and a 118 bp mutant band. |

*Continued*

| Reagent type (species) or resource | Designation | Source or reference | Identifiers | Additional information |
|---|---|---|---|---|
| Sequence-based reagent | Ift122(TR2)_R | This study | PCR primer | ACTCCCAAGC AAGCGAACT |
| Sequence-based reagent | Ttc21b(TF2)_F | This study | PCR primer | AGAATGATGTGC AACCTTGTTGA After amplification with below reverse primer, product is digested with NmuCl, resulting in a 224 bp WT band and a 168 bp mutant band. |
| Sequence-based reagent | Ttc21b(TF2)_R | This study | PCR primer | TTATCTGGCTCA CGGTCTCC |
| Software, algorithm | SeedWater Segmenter | *Mashburn et al., 2012* | | |
| Software, algorithm | SEGGA | *Farrell et al., 2017* | | |
| Software, algorithm | FIJI/ImageJ | *Schindelin et al., 2012 Schneider et al., 2012* | RRID:SCR_002285 | |
| Software, algorithm | MorphoLibJ (FIJI plugin) | *Legland et al., 2016* | | |
| Software, algorithm | ITK-SNAP | *Yushkevich et al., 2006* | RRID:SCR_002010 | |
| Software, algorithm | R | *R Development Core Team, 2020* | RRID:SCR_001905 | |
| Software, algorithm | Circular plugin (for R) | *Agostinelli and Lund, 2017* | | |
| Software, algorithm | Prism | Graphpad | RRID:SCR_002798 | |
| Software, algorithm | Zen | Zeiss | RRID:SCR_018163 | |
| Software, algorithm | LAS X | Leica | RRID:SCR_013673 | |
| Software, algorithm | EOS Utility | Canon | | |
| Software, algorithm | Illustrator | Adobe | RRID:SCR_010279 | |

## Mouse strains

The *Ift122*$^{TR2}$ and *Ttc21b*$^{TF2}$ alleles were identified in an ongoing forward genetic screen (*García-García et al., 2005*). The *Ift122*$^{TR2}$ mutation was mapped to a single C to A mutation at position 115,899,529 on chromosome 6, resulting in a premature stop codon that is predicted to truncate the protein at amino acid 575 (out of 1138). The *Ttc21b*$^{TF2}$ allele was mapped to a single C to A mutation at position 66,242,780 on chromosome 2, resulting in a premature stop codon that is predicted to truncate the protein at amino acid position 187 (out of 1,315). Both alleles created a new restriction site. The presence of the *Ift122*$^{TR2}$ allele was genotyped by PCR amplification with the primers TR2F 5' CTGGTTGTAATCTGACTCGTTGA 3' and TR2R 5' ACTCCCAAGCAAGCGAACT 3' followed by restriction digest with HpyCH4III (New England Biolabs). The presence of the *Ttc21b*$^{TF2}$ allele was genotyped by PCR amplification with the primers TF2F 5' AGAATGATGTGCAACCTTGTTGA 3' and TF2R 5' TTATCTGGCTCACGGTCTCC 3' followed by restriction digest with NmuCl (ThermoFisher Scientific). The following previously described mouse strains were used in this study: Wnt1-Cre2 [Tg(Wnt1-cre)2Sor] (*Lewis et al., 2013*), EIIA-Cre [Tg(EIIa-Cre)C5379Lmgd/J] (*Lakso et al., 1996*), mT/mG [Gt(ROSA)26Sortm4(ACTB-tdTomato,-EGFP)Luo/J] (*Muzumdar et al., 2007*), SmoM2 [Gt(ROSA)26Sortm1(Smo/YFP)Amc/J] (*Jeong et al., 2004*), and Gli2 [Gli2$^{tm2.1Alj}$/J] (*Bai and Joyner, 2001*). All lines were maintained on an FVB/N background except SmoM2, which was maintained on a C57BL/6J background, and *Gli2*, which was maintained on an SWR/J background. Timed pregnant mice were euthanized at E7.5-E12.5. Noon on the day of the vaginal plug was considered E0.5 and embryos were staged by counting the number of somites. Analysis of wild-type embryos in *Figures 1–3* and associated supplements was performed on FVB/N embryos. Control embryos were wild-type and heterozygous littermate controls of *Ift122*, *Ttc21b*, and *Gli2*

mutants (designated WT in the corresponding figures), or embryos bearing Wnt1-Cre2 or SmoM2 alone (designated control in *Figure 9* and *Figure 9—figure supplements 1* and *2*). The presence or absence of exencephaly was analyzed in E10.5-E12.5 embryos. Mutant and transgenic embryos were processed in parallel with littermate controls.

## Whole-mount immunostaining

Embryos were dissected in ice-cold PBS and fixed overnight at 4°C in 4–8% paraformaldehyde (PFA, Electron Microscopy Sciences) or Dent's fixative (4:1 methanol:DMSO). Embryos fixed in Dent's fixative were rehydrated in successive 30 min washes of 75:25, 50:50, and 25:75 methanol:PBS at room temperature (RT). Rehydrated embryos were then washed 3 × 30 min in PBS + 0.1% TritonX100 (PBTriton) at RT. Embryos were then incubated in blocking solution (PBS + 3% BSA, 0.1% TritonX100) for 1 hr at room temperature. Embryos were then incubated in staining solution (PBS + 1.5% BSA, 0.1% TritonX100) containing primary antibodies overnight at 4°C. Embryos were then washed 3 × 30 min in PBTriton and incubated in staining solution containing Alexa Fluor conjugated secondary antibodies (1:500, ThermoFisher) for 1 hr at room temperature. Embryos were subsequently washed 3 × 30 min in PBTriton at RT and stored in PBTriton at 4°C until imaging. Antibodies used for embryos fixed in Dent's fixative were: rat anti-ZO-1 (DSHB R26.4C, 1:100), rabbit anti-Arl13b (*Caspary et al., 2007*) (1:1000), rabbit anti-phosphohistone H3 (Upstate 06–570, 1:1000), and mouse anti-β-catenin (BD Biosciences 610153, 1:300). Antibodies used for embryos fixed in 4% PFA were rabbit anti-laminin (Sigma L9393, 1:1000), rabbit anti-N-cadherin (Cell Signaling Technology D4R1H, 1:500), chicken anti-GFP (abcam ab13970, 1:1000), mouse anti-β-catenin (BD Biosciences 610153, 1:300), mouse anti-Nkx6.1 (DSHB F55A10, 1:50), and rabbit anti-FoxA2 (abcam ab108422, 1:1000). Embryos fixed in 8% PFA were stained with rabbit anti-diphosphomyosin regulatory light chain antibody (Cell Signaling Technology 3674, 1:100). Alexa 546-conjugated phalloidin (Molecular Probes), and DAPI (ThermoFisher) were used as counterstains.

## Cryosectioning

Embryos were dissected and fixed in 4% PFA for 2 hr at room temperature and then washed 5 × 30 min in PBTriton. Embryos were then transferred into 15% sucrose for 30 min and subsequently into 30% sucrose overnight at 4°C. Embryos were then placed anterior down in a cryoblock in OCT (Tissue-Tek) and frozen on dry ice. Embryos were stored at −80°C until sectioning. Embryos were sectioned on a cryostat (Leica) from anterior to posterior in 14 μm sections, with sections adsorbed onto Superfrost slides (Fisher). Cryosections from the midbrain/hindbrain region were then washed 3 × 15 min in PBTriton at RT, blocked for 30 min in blocking solution (see above), stained for 30 min with primary antibodies as above, washed 3 × 15 min in PBTriton, incubated with secondary antibodies, and washed 3 × 15 min in PBTriton. Stained sections were then mounted under a coverglass in fluorescence mounting media (Dako).

## Microscopy

For whole-mount confocal analysis, stained embryos were mounted dorsal side down in PBTriton in Attofluor cell chambers (ThermoFisher A7816), using a small fragment of broken coverglass with small dabs of vacuum grease (Dow Corning) to mount the embryo on a #1.5 coverglass (Dow Corning). Embryos were then imaged by inverted confocal microscopy on either a Zeiss LSM700 equipped with a Plan-NeoFluar 40x/1.3 oil immersion objective, or a Leica SP8 equipped with a HC PL Apo 40x/1.3 oil immersion objective. Images were captured by tile-based acquisition of contiguous z-stacks of 50–150 μm depth with 0.9–1.2 μm optical slices and 0.3–0.5 μm z-steps. Tiled images were computationally stitched together with 10% overlap per tile using Zen (Zeiss) or LAS-X (Leica) software, resulting in visible seams in some images. Maximum-intensity projections of the entire z depth were created for analysis in the same software. For confocal imaging of cryosections, slides were imaged on an inverted Zeiss LSM700 equipped with a Plan-Apochromat 20x/0.8 air objective. Z-stacks of 10–14 μm depth were imaged with 1.8–2.0 μm optical slices and 1.0–1.2 μm z-steps. For bright-field imaging, embryos were imaged in PBTriton on a Zeiss Stemi 508 stereomicroscope equipped with a Canon EOS DSLR camera and EOS Utility software (Canon).

## Image analysis and quantification

Apical area was measured in 100 μm x 100 μm regions in maximum-intensity projections of tiled images, either at the midline or in a pair of regions on either side of the midline, approximately midway between the midline and the lateral extent of the neural plate and midway between the pre-otic sulcus and the cranial flexure. For the analysis of cell area throughout the mediolateral axis, a continuous series of 100 μm (anterior-posterior) x 20 μm (mediolateral) regions from the midline to the lateral edge were analyzed. Cells contained entirely within these regions were segmented using SeedWater Segmenter software (*Mashburn et al., 2012*). Cell areas were quantified and area maps were generated using the MorphoLibJ plugin (*Legland et al., 2016*) in the FIJI redistribution of ImageJ (*Schindelin et al., 2012*; *Schneider et al., 2012*). Cell orientation was evaluated in the same regions using SEGGA software (*Farrell et al., 2017*). Cells were assigned a mediolateral orientation if they were oriented at 0–45° with respect to the mediolateral axis or an anterior-posterior orientation if they were oriented at 45–90°. Cell height was measured in cryosections by drawing a perpendicular line in FIJI from the apical to the basal surface between two apparent cell edges using phalloidin and laminin or β-catenin (*Figure 3C–E*), or in XZ-reconstructions of embryos stained with phalloidin (*Figure 8G and H*). The ratio of the apical span to the basal span of the tissue was calculated by manually drawing segmented lines in FIJI from one lateral extreme of the neural plate to the other in cryosections.

Three-dimensional lateral cell shapes were analyzed using the 3D Project tool in FIJI in cells labeled by EIIA-Cre-driven mosaic recombination of the mT/mG locus, which frequently resulted in individually labeled cells. Cells throughout the lateral midbrain region were analyzed and were manually assigned to shape categories based on examination of their apical, mid-, and basal cross-sectional areas. Cells were considered apically constricted if their apical surface was smaller than their basal surface. EIIA-Cre produced little to no labeling in the midline, and midline cell 3D analysis was performed by manual segmentation of cells labeled with β-catenin using ITK-SNAP software (*Yushkevich et al., 2006*). Comparison of apical and basal areas of midline cells was performed in FIJI.

For analysis of cell proliferation, the percentage of phosphohistone H3-positive cells was calculated in contiguous 100 μm x 100 μm regions along the mediolateral axis (*Figure 4—figure supplement 5*) or in a single 100 μm x 100 μm region midway between the midline and the lateral edge of the tissue (*Figure 9—figure supplement 1*). Mesenchymal cell density was calculated by counting the number of individual DAPI-labeled nuclei in a 50 μm x 100 μm region in transverse sections.

The number and angle of multicellular F-actin and phosphomyosin cables were analyzed manually using FIJI in a pair of 100 μm x 100 μm lateral regions on either side of the midline in each embryo. A cable was defined as three or more consecutive edges of high-intensity signal with no gap or diminishment along its length. Apical F-actin and phosphomyosin intensity were analyzed in 50 cells in a 50 μm x 50 μm lateral region in embryos stained for phalloidin (F-actin) and phosphorylated myosin II by calculating the mean intensity of a line drawn along the entire apical cortex of each cell. All intensity quantifications were performed on unprocessed maximum intensity projections.

## Statistics and figure assembly

Statistical analyses and graph generation were performed in Prism software (Graphpad) or with the circular plugin (*Agostinelli and Lund, 2017*) in the R software package (*R Development Core Team, 2020*). All results are reported as mean ± standard deviation (SD). Summary significance levels are as follows: ***$p<0.001$, **$p<0.01$, *$p<0.05$. Statistical tests were Welch's t-test, which does not assume equal SDs between conditions, the Kolmogorov-Smirnov test for comparing distributions, the standard one-way ANOVA with Tukey's multiple comparisons, which was used when the variance between replicates was expected to come only from measurement error, the Brown-Forsythe and Welch one-way ANOVA using Dunnett's T3 multiple comparisons test, which does not assume equal SDs between conditions, the two-way ANOVA with Sidak's multiple comparisons for comparing multiple conditions, and the Watson nonparametric two-sample test for homogeneity for examining circular distributions. Details of the statistical tests, n values, and p values for each experiment can be found in *Supplementary file 1*. Figures were assembled using Photoshop and Illustrator (Adobe). For display purposes, some plots did not show cells outside the x-axis range, which were generally <2% of cells, except in *Figure 5D* (<4% of cells excluded) and *Figure 9K* (<8% of cells

excluded). All cells were included in the statistical analysis. Each embryo was considered a biological replicate. Formal power analyses were not conducted. For mutant analyses, in which mutant embryos were compared with stage-matched littermate controls, an n of three to five embryos per genotype was targeted.

## Acknowledgements

The authors thank Heather Alcorn for identifying and mapping the *Ift122^TR2* and *Ttc21b^TF2* alleles, Alex Joyner for helpful discussions and for the *Gli2^lzki* mice, Ann Sutherland for introducing ERB and JAZ to the mouse neural plate, Ian Prudhomme for technical assistance, and Marissa Gredler, Matthew Schilling, Masako Tamada, and Richard Zallen for comments on the manuscript. This work was supported by NIH/NINDS F32 fellowship NS098832 to ERB and MSKCC Cancer Center Support Grant P30 CA008748. JAZ is an investigator of the Howard Hughes Medical Institute.

## Additional information

### Funding

| Funder | Grant reference number | Author |
|---|---|---|
| Howard Hughes Medical Institute | | Jennifer A Zallen |
| National Institutes of Health | F32 NS098832 | Eric R Brooks |
| National Institutes of Health | P30 CA008748 | Jennifer A Zallen |

The funders had no role in study design, data collection and interpretation, or the decision to submit the work for publication.

### Author contributions

Eric R Brooks, Conceptualization, Formal analysis, Funding acquisition, Investigation, Writing - original draft, Writing - review and editing; Mohammed Tarek Islam, Investigation, Writing - review and editing; Kathryn V Anderson, Conceptualization, Resources, Writing - review and editing; Jennifer A Zallen, Conceptualization, Resources, Formal analysis, Supervision, Funding acquisition, Writing - review and editing

### Author ORCIDs

Eric R Brooks http://orcid.org/0000-0003-3159-8626
Jennifer A Zallen https://orcid.org/0000-0003-3975-1568

### Ethics

Animal experimentation: All animal experiments were conducted in accordance with the Guide for the Care and Use of Laboratory Animals of the National Institutes of Health and an approved Institutional Animal Care and Use Committee protocol (15-08-13) of Memorial Sloan Kettering Cancer Center.

### Decision letter and Author response

Decision letter https://doi.org/10.7554/eLife.60234.sa1
Author response https://doi.org/10.7554/eLife.60234.sa2

## Additional files

### Supplementary files

• Supplementary file 1. N values and details of statistical analyses performed. The details of the statistical tests performed, including the exact n and p values in the Figures and Figure supplements, are presented.

- Transparent reporting form

## Data availability

All data generated or analyzed during this study are included in the manuscript and supporting files.

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
