## [Decision Letter]

**Acceptance summary:**

Your work identifies novel cell biological and developmental underpinnings of midbrain exencephaly by identifying how cilium and Hh pathway activity changes apical constriction and apicobasal lengthening in the lateral neural tube. Highlights of your work include the compelling imaging and careful quantitation of cell dimensions. We appreciate the extensive attention to the revision process.

**Decision letter after peer review:**

Thank you for submitting your article "Sonic hedgehog directs patterned apical constriction during mammalian cranial neural tube closure" for consideration by *eLife*. Your article has been reviewed by three peer reviewers, one of whom is a member of our Board of Reviewing Editors, and the evaluation has been overseen by Marianne Bronner as the Senior Editor. The reviewers have opted to remain anonymous.

The reviewers have discussed the reviews with one another and the Reviewing Editor has drafted this decision to help you prepare a revised submission.

Neural closure defects represent common profound birth defects. Your work provides an assessment of apical constriction during cranial (midbrain) neural closure in the mouse embryo. Although genes that regulate apical constriction are known to result in neural tube defects, the cellular-level consequences have not been well-studied. You reveal a medial-lateral pattern to apical constriction as neurulation progresses. You manipulate cilia/Shh signaling and assess changes in apical constriction and neuroepithelial morphology, relating this to neural closure defects. The experiments are carefully performed and the imaging is compelling. The concept of apical constriction in a broad region, rather than solely at "hinge points", has been raised previously, and the current studies provide quantitative assessment as well as some molecular mechanistic insights. The reviewers thought the strengths of the paper were extensive, but thought that the mechanistic insights into which cells were driving the closure, and how Hedgehog signaling may be directing this cell behavior were limited. As described below, the reviewers ask for a more careful analysis of the behavior of the midline cells and extending the apical constriction analysis to a model of a low Gli activity state, as described below.

Essential revisions:

1) As shown in Figure 2, when taken as an aggregate, midline cells do not change their average apical surface area significantly. However, their planar orientation appears to change quite dramatically over the stages examined (as implied in Figure 1C as well). The authors should analyze and quantify this orientation pattern, as well as include analysis of the midline cells in the Ift-A mutants (in part shown in Figure 5 but not analyzed), which show altered Shh signaling. If planar orientation is affected, how might this effect influence the interpretation of the remainder of the data?

2) The authors have only studied mutants that result in ligand-independent activation of the pathway (leading to ectopic gain of function of Gli activity). Previously, the authors have analyzed IFT-B mutants and other mutants with Gli decreased activity. One or more mutants should be studied as well to assess the phenotype of Gli decreased activity in both midline and lateral cells to better tease out the function of HH signaling in directing apical area.

---

## [Author Response]

Essential revisions:1) As shown in Figure 2, when taken as an aggregate, midline cells do not change their average apical surface area significantly. However, their planar orientation appears to change quite dramatically over the stages examined (as implied in Figure 1C as well). The authors should analyze and quantify this orientation pattern, as well as include analysis of the midline cells in the Ift-A mutants (in part shown in Figure 5 but not analyzed), which show altered Shh signaling. If planar orientation is affected, how might this effect influence the interpretation of the remainder of the data?

To address this question, we analyzed cell orientation at different stages of elevation in both midline and lateral regions. We observe an increase in mediolateral cell elongation and the percentage of mediolaterally aligned cells in the lateral region during elevation (Figure 2—figure supplement 1A-C). Midline cell orientation did not change significantly during elevation (Figure 2—figure supplement 1D-F). These changes in cell shape are not predicted to cause the observed decrease in mediolateral width of the apical neural plate (Figure 3—figure supplement 1A), and therefore do not change our interpretation that apical constriction is the primary contributor to cranial neural fold elevation. We note that fluctuations in local cell alignment can create the appearance of differences between different fields of view. To better convey the trends we observe, we now show all of the images used to analyze midline cell morphology and one of two images/embryo used to analyze lateral cell morphology in wild type (new Figure 2—figure supplement 1A and D).

We also extended this analysis to mutant backgrounds. Mediolateral cell orientation was slightly, but significantly, decreased in lateral cells of *Ift122* and *Ttc21b* mutants (Figure 4—figure supplement 4), but not in *Gli2* mutant or SmoM2-expressing embryos (Figure 8—figure supplement 2, Figure 9—figure supplement 2). Midline cell orientation was unaffected in all backgrounds. These results indicate that changes in cell orientation do not make a major contribution to the mutant phenotypes we observe.

2) The authors have only studied mutants that result in ligand-independent activation of the pathway (leading to ectopic gain of function of Gli activity). Previously, the authors have analyzed IFT-B mutants and other mutants with Gli decreased activity. One or more mutants should be studied as well to assess the phenotype of Gli decreased activity in both midline and lateral cells to better tease out the function of HH signaling in directing apical area.

Thank you for this important suggestion. To address this question, we added a new analysis of cell morphology in mutants lacking *Gli2*, the predominant positive effector of Shh-dependent transcription in the cranial neural plate. We found that lateral cells in *Gli2* mutants had no defects in apical area or height, suggesting that Shh signaling through *Gli2* is not required for lateral cell architecture (Figure 8 and Figure 8—figure supplements 1 and 2). By contrast, midline cells in *Gli2* mutants showed increased cell height and decreased apical area, similar to the defects in midline cells of *Ift122* and *Ttc21b* mutants. These results suggest that Shh signaling mediated by *Gli2* is required for the short, apically expanded morphology of midline cells but is dispensable for lateral cell architecture. Because *Gli2* mutants complete neural tube closure (Mo et al., 1997; Matise et al., 1998; Bai et al., 2002), these results provide further support to our conclusion that lateral (and not midline) cells are the main drivers of cranial neural fold elevation.